



# 1 Is the Atlantic Ocean driving the recent variability in South
# 2 Asian dust?

Priyanka Banerjee[1], Sreedharan Krishnakumari Satheesh[1,2], Krishnaswamy Krishna Moorthy[2]
[1]Divecha Centre for Climate Change, Indian Institute of Science, Bangalore, India
[2]Centre for Atmospheric and Oceanic Sciences, Indian Institute of Science, Bangalore, India
*Correspondence to*: Priyanka Banerjee (pbanerjee.ocean@gmail.com)
**Abstract**
This study investigates the large-scale factors controlling interannual variability of dust aerosols over South
Asia during 2001-2018. We use a parameter $DA_\%$, which refers to the frequency of days in a year when high
dust activity is experienced over a region, as determined by combination of satellite aerosol optical depth and
Angstrom exponent. While positive sea surface temperature (SST) anomaly in the central Pacific Ocean has
been important in controlling $DA_\%$ over South Asia during 2001-2010; in recent years the North Atlantic Ocean
has assumed a dominant role. Specifically, high $DA_\%$ is associated with warming in the mid-latitude and cooling
in the sub-tropical North Atlantic SSTs: the two southern arms of the North Atlantic SST tripole pattern. This
shift towards a dominant role of the North Atlantic SST in controlling $DA_\%$ over South Asia is associated with a
recent shift towards persistently positive phase of the North Atlantic Oscillation (NAO) and a resultant positive
phase of the spring-time SST tripole pattern. Interestingly, there has also been a shift in the relation between the
two southern arms of the SST tripole and NAO, which has resulted in weakening of the southwest monsoon
circulation over the northern Indian Ocean and strengthening of the dust-carrying westerlies and northerlies in
the lower and mid-troposphere. Simulations with an earth system model show that anomalous transport due to
the North Atlantic SST tripole pattern can result in 10% (20%) increase in dust optical depth (concentration at
800 hPa) over South Asia during May-September; with increases as much as 30% (50%) during the month of
June.
**1      Introduction**
South Asia is believed to be highly vulnerable to the long-term impacts of climate change (Stocker et al., 2013).
One of the ways in which the impact of climate change is felt in this region is via aerosol feedback on the
regional climate (e.g., Satheesh and Ramanathan, 2000; Ramanathan et al., 2005; Bollasina et al., 2011).
Mineral dust is the most important aerosol component (by mass) present in this region (e.g., Ginoux et al., 2012;
Jin et al., 2018a; Banerjee et al., 2019). Several studies during the last two decades have shown that mineral dust
can influence different aspects of the climate of South Asia with the largest focus given to dust impact on
radiative balance (e.g., Deepshikha et al., 2006; Zhu et al., 2007; Pandithurai et al., 2008) and the southwest
monsoon (SWM) precipitation (Vinoj et al., 2014; Jin et al., 2014; Solmon et al., 2015). However, to better
appreciate dust-climate feedback, it is important to understand what large-scale factors control dust emission



and transport in this region and, if there are long-term changes in these controlling factors. At present, there is
very little understanding of these factors, sometimes with lack of consensus among the studies.
There are some recent indirect evidences of El Nino/La Nina influencing dust fluxes over South Asia. For
example, Kim et al. (2016) have reported that La Nina conditions are associated with increased absorbing
aerosols over northwest India which, in turn, leads to positive feedback on the SWM precipitation. On the
contrary, Abish and Mohankumar (2013) argued that increased zonal transport and subsidence over India during
El Nino years can lead to enhanced absorbing aerosols like dust over India. A few other studies have shown that
over Southwest Asia, variability of dust aerosols is controlled by climatic factors like El Nino/La Nina at
interannual timescale (Notaro et al., 2015; Yu et al., 2015; Banerjee and Prasanna Kumar, 2016) and by Pacific
Decadal Oscillation (PDO) at interdecadal timescale (Notaro et al., 2015; Yu et al., 2015; Pu and Ginoux, 2016).
Eastward transport of dust from Southwest Asia by the mid-level westerlies are shown to contribute about 50%
to the total dust optical depth over the Indo-Gangetic plain of South Asia (Banerjee et al., 2019) and can
influence its trend over this region. During the beginning of the 21$^{st}$ century, a positive trend in SWM
precipitation due to the negative phase of Interdecadal Pacific Oscillation (Huang et al., 2020) has resulted in a
negative trend of dust aerosol over South Asia (Pandey et al., 2017; Jin and Wang, 2018b). Ice core records in
the central Himalayas have shown an inverse relation between the SWM precipitation and dust deposition
(Thompson et al., 2000). During winter season, aerosol optical depth over northern India is shown to be
positively correlated to simultaneous central Pacific Nino index and negatively correlated to Antarctic
Oscillation during the preceding autumn (Gao et al., 2019).
The main dust source regions over South Asia are spread across the Thar Desert and the Indo-Gangetic plain in
India and Pakistan; the Makran coast and the Hamun-I-Mashkel in Pakistan; the Margo Desert and the Rigestan
Desert in Afghanistan (Walker et al., 2009; Ginoux et al., 2012). The Margo Desert, the Rigestan Desert and the
Hamun-I-Mashkel receive predominantly winter precipitation from the Mediterranean low-pressure systems
travelling eastwards. Rest of the regions receive summer precipitation from the SWM system, although the total
amount of precipitation received is very low. It has been shown by several studies that one of the major factors
controlling the interannual variability of the SWM rainfall is El Nino/La Nina with developing El Nino
conditions over the Pacific Ocean leading to weakening of the SWM moisture influx (e.g., Sikka, 1980;
Rasmusson and Carpenter, 1983; Ashok et al., 2004). Tropical Pacific Ocean warming (cooling) is also
responsible for wetter (drier) than normal conditions over the winter precipitation region in Southwest Asia
(Barlow et al., 2002; Mariotti, 2002). This implies that the conditions prevailing over the Pacific Ocean has an
important role in controlling the level of dust activity over the northern Indian Ocean (IO) and South Asia either
directly through precipitation impact on dust emission and/or indirectly through dust transport from Southwest
Asia. However, in the backdrop of global warming and the internal variability of the Pacific Ocean at different
timescales (e.g., Kosaka and Xie, 2016; Deser et al., 2017a), the well-known El Nino-monsoon relation has
undergone changes in the recent decades. Since the late 1970s, the relation between El Nino and negative
rainfall anomaly over India has become less significant, possibly, due to the greater rate of warming of the
Eurasian landmass in the recent years compared to the IO or due to the cooling of the Pacific Ocean (Kumar et
al., 1999; Kinter et al., 2002). Simultaneously, the Atlantic Ocean has assumed a stronger role in modulating the
monsoon circulation over the northern IO (Chang et al., 2001; Kucharski et al., 2007; Kucharski et al., 2008).



While some studies have shown the importance of the sea surface temperature (SST) along the south equatorial
Atlantic (Kucharski et al., 2007; Kucharski et al., 2008), other studies have shown that positive SST anomalies
over the western North Atlantic centered on 40$^o$N latitude can lead to positive anomalies of monsoon over India
(Srivastava et al., 2002; Rajeevan and Sridhar, 2008). Over the North Atlantic Ocean, the dominant mode of sea
level pressure variability during winter is the North Atlantic Oscillation (NAO) (Hurrell, 1995). The tripole
pattern of SST over the North Atlantic associated with the winter NAO (see e.g., Visbeck et al., 1998) can
persist during spring and impact the summer circulation over Eurasia (Gastineau and Frankignoul, 2015; Osso et
al., 2018). During summer months, two dominant modes of variability are the summer NAO (Folland et al.,
2009) and the Summer East Atlantic (SEA) pattern (Osso et al., 2018; Osso et al., 2020). During the period
1948-2016, for the summer months of June-September, NAO explained about 36% of variance, while SEA
explained about 16% of variance of sea level pressure (Osborne et al., 2020). A few studies have shown that
such variability of SST and circulation over the North Atlantic has the potential to influence the SWM
circulations over South Asia. For example, the SST anomalies associated with the Atlantic Multidecadal
Oscillations can influence the tropospheric temperature leading to strengthening or weakening of the monsoon
via modulation of the frequency and strength of NAO (Goswami et al., 2006). The cold (positive) phases of the
SST tripole over the North Atlantic have induced stronger westerlies over the northern IO (Krishnamurthy and
Krishnamurthy, 2015). The influences of the extra-tropical North Atlantic/Pacific SST on the South Asian
monsoon are stronger during weak El Nino/La Nina years (Chattopadhyay et al., 2015).
In the above backdrop, we examine how changes in the spatial pattern of ocean warming during 2001-2018 have
led to increased dependence of South Asian dust on the North Atlantic Ocean, shifting from the previously
dominant influence of the equatorial Pacific SST. Using observations and reanalysis data we explore the
physical mechanism by which a remote response of the circulation over South Asia is invoked by SST
anomalies over the North Atlantic. We have further performed control and sensitivity studies using an earth
system model to investigate in detail how dust emission and transport is impacted by perturbing SST over the
North Atlantic Ocean. For this study, we have chosen a domain encompassing 65$^o$E-82$^o$E longitude and 24$^o$N-
32$^o$N latitude. We consider this as the dust belt of South Asia. The region is influenced predominantly by SWM
precipitation. Unless stated otherwise, all analyses involving spatial averaging focus only on this region.

**2      Data and Models**
**2.1     Satellite observation and reanalysis data**
The main source of dust aerosol data for this study is from the Moderate Resolution Imaging Spectroradiometer
(MODIS) aboard Terra (2001-2018) and Aqua (2003-2018) satellites, which provide the longest satellite-based
information on both aerosol load and size distribution over land and ocean. We have calculated frequency of
days in a year when substantial dust activity is experienced over South Asia (DA$_\%$) using MODIS level 3
version 6.1 daily deep blue aerosol optical depth ($\tau$) and Angstrom exponent ($\alpha$). The deep blue algorithm of
MODIS is used to retrieve aerosol information over bright surfaces, like arid regions, where surface reflectance
is low at the blue end of the spectrum (Hsu et al., 2004; Hsu et al., 2006). The criteria used for estimating DA$_\%$




are (i) Ṭ > 0.6 and (ii) α < 0.2 to isolate the days dominated by moderately high load of coarse-mode aerosols.
This yields a map of the main dust source regions in and around South Asia at $1^o$X$1^o$ horizontal resolution.
Previously, along with deep blue Ṭ and α, single scattering albedo has also been used to account for the
absorptive property of dust when deriving dust optical depth (Ginoux et al., 2012; Pu and Ginoux, 2018). For
our present purpose, Ṭ and α combination is sufficient since we are deriving frequency of days of dust activity
and not the absolute optical depth. Fig. 1a shows the spatial distribution of DA$_\%$ averaged for 2001-2018 and its
standard deviation (SD). High values of DA$_\%$ coincide with known locations of dust source regions. The SD is
low indicating that high dust activities persist over these regions. The inset in Fig. 1a shows the monthly
climatology of DA$_\%$ with the SD, which reveals that highest values occur during June-July and lowest values
during November. Over the dust belt of South Asia, for 2001-2018, average DA$_\%$ from MODIS Terra is 5.2 (SD
is 1.7) and from MODIS Aqua is 4.2 (SD is 1.7). Changing the threshold values of both Ṭ and α by 50% and
recalculation of DA$_\%$ does not lead to any significant changes in these results. MODIS-derived DA$_\%$ matches
well with year-to-year variability of dust optical depth (Ṭ$_d$) from Infrared Atmospheric Sounder Interferometer
(IASI) aboard Metop-A (2008-2018) with a correlation coefficient of 0.73, which is significant at 99%
confidence level (Fig. 1b). IASI reports Ṭ$_d$ at 10 µm wavelength and at a spatial resolution of $0.5^o$X$0.5^o$ (Capelle
et al., 2018). For 2008-2018, IASI dataset yields annual average Ṭ$_d$ value of 0.17 (SD of 0.02). In subsequent
analysis, we use combined DA$_\%$ obtained from MODIS Terra and Aqua.

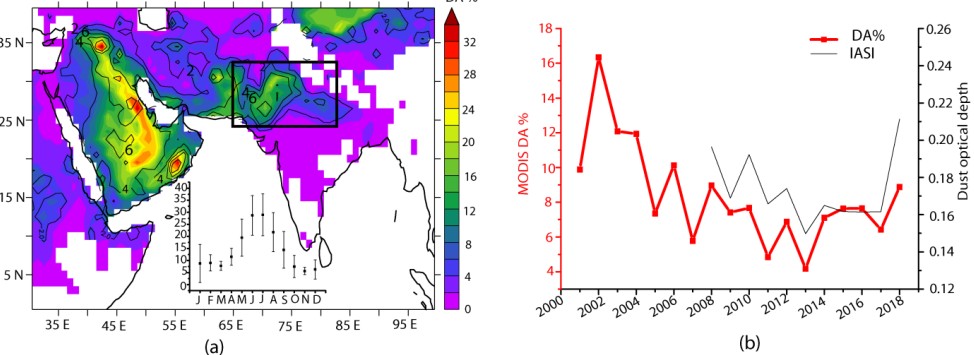

(a)    (b)
**Figure 1: (a) Shading shows spatial distribution of DA$_\%$ averaged for 2001-2018 and contours are the standard**
**deviations of DA$_\%$ for the same period. The black rectangle indicates the dust belt of South Asia ($65^o$E-$82^o$E, $24^o$N-**
**$32^o$N) which is used for subsequent analysis. The monthly climatology and the standard deviation of DA$_\%$ over dust**
**belt of South Asia are shown by black squares and vertical bars respectively in the inset. (b) Time-series of MODIS-**
**derived DA$_\%$ and IASI-retrieved annual dust optical depth over South Asia.**
To examine the linkages between the spatial variability of SST during different periods and South Asian dust
activity, we have used 3 SST datasets: (1) National Oceanic and Atmospheric Administration (NOAA)
Extended Reconstructed SST (ERSST) version 5 (Huang et al., 2017) available at $2^o$X$2^o$ spatial resolution, (2)
Centennial in situ Observation-Based Estimates (COBE) version 2 SST data at $1^o$X$1^o$ spatial resolution
(Hirahara et al., 2014) and (3) Optimally Interpolated SST version 2 (OISST) data at $1^o$X$1^o$ spatial resolution



(Reynolds et al., 2002). All the SST datasets are at monthly temporal resolution. The ERSST version 5 data
combines ship and buoy SST from International Comprehensive Ocean and Atmosphere Dataset (ICOADS)
along with Argo data since 2000. COBE also uses ICOADS data along with data from Kobe collection. Finally,
OISST combines Advanced Very High Resolution Radiometer (AVHRR) retrievals with ship-borne and buoy
data. Atmospheric data such as wind vectors, geopotential height, sea level pressure and velocity potential have
been taken from National Centers for Environmental Prediction/ National Center for Atmospheric Research
(NCEP/NCAR) Reanalysis at $2.5^{o}X2.5^{o}$ spatial resolution (Kalnay et al., 1996). For precipitation we have used
monthly Global Precipitation Climatology Project (GPCP) version 2.3 data available at $2.5^{o}X2.5^{o}$ spatial
resolution, which combines rain gauge measurements with satellite observations (Huffman et al., 1997).
Additionally, monthly precipitation data averaged from daily data has been obtained from Precipitation
Estimation from Remotely Sensed Information using Artificial Neural Networks (PERSIANN) at $0.25^{o}X0.25^{o}$
spatial resolution. PERSIANN algorithm is applied on Gridded Satellite (GridSat-B1) brightness temperature
observation in the infrared region (Ashouri et al., 2015). The precipitation data are then corrected for bias
against GPCP precipitation estimates. To track the large-scale variability over North Atlantic, Hurrell's station-
based seasonal NAO index has been used for the years 2001-2018 (Hurrell, 1995; Hurrell and Deser, 2009).
NAO index is calculated based on the difference between normalized sea level pressure over Lisbon, Portugal
and Stykkisholmur/Reykjavik, Iceland.
**2.2    CESM experiments**
Simulations were carried out using the Community Earth System Model (CESM) version 1.2 to examine the
mechanism by which SST anomalies over the North Atlantic Ocean impact dust cycle over South Asia. CESM
is a fully coupled model used for simulations of global climate across different spatial and temporal scales.
There are several components to CESM model (example atmosphere, land, sea ice, ocean etc.), which are linked
through a coupler. We have used Community Atmosphere Model version 4 with the Bulk Aerosol Module
(CAM4-BAM) coupled with Community Land Model version 4 in "Satellite Phenology" (CLM-SP)
configuration. Simulations are carried out for trace gases levels corresponding to the year 2000 at $0.9^{o}X1.25^{o}$
spatial resolution with 26 levels in the vertical.
Emission of dust is calculated within CLM model, while dust transport and deposition as well as the radiative
effects are calculated within CAM model (Mahowald et al., 2006). Dust emission follows the treatment of Dust
Entrainment and Deposition scheme of Zender et al. (2003a). Dust emission is based on saltation process, which
depends on modelled wind friction velocity, soil moisture, vegetation and snow cover. This saltation flux occurs
whenever wind friction velocity exceeds a threshold (Marticorena and Bergametti, 1995). Additionally, dust
emission is corrected by a geomorphic source function, which accounts for the spatial variability of erodible
materials (Zender et al., 2003b). In CAM4-BAM dust is emitted in 4 size bins: 0.1-1.0, 1.0-2.5, 2.5-5.0 and 5.0-
10.0 µm. Dust is transported based on CAM4 tracer advection scheme and is removed via dry (gravitational and
turbulent deposition) and wet depositions (convective and large-scale precipitation) (Zender et al., 2003a; Neale
et al., 2010). The solubility factor and scavenging coefficient are taken here as 0.15 and 0.1 respectively.
Two sets of simulations have been carried out with CESM: (1) the "Ctrl" simulation, where the atmosphere was
forced with prescribed climatological monthly SST and sea ice from Hadley Centre (1870-1981) and NOAA



Optimal Interpolation SST (1981-2010) (Hurrell et al., 2008), and (2) the "NAtl" simulation, where the month-
by-month observed trend in SST during 2011-2018 were imposed over the climatological SST only over the
North Atlantic Ocean. Over rest of the domain climatological SST from Hurrell was prescribed. Thus, the
differences between "NAtl" and "Ctrl" simulations reflect solely the impact of North Atlantic SST anomalies, as
observed during 2011-2018, on atmospheric circulation and dust load. A total of 15 years of simulations have
been carried out for each of Ctrl and NAtl cases with each year being initialized from the atmospheric state at
the end of the previous year. For this study, monthly mean values for the last 10 years of model runs have been
used for both the cases. We have compared $\tau_d$ from Ctrl run with IASI-retrieved $\tau_d$ and coarse mode $\tau$ data
from Aerosol Robotic Network (AERONET) stations at Kanpur (2001-2018), Lahore (2010-2016) and Jaipur
(2010-2017). For this, we have used version 3 AERONET level 2.0 cloud cleared aerosol data.

**3      Results and discussion**
We first demonstrate that there is a change in the relation between dust aerosol variability over South Asia and
global SSTs during 2001-2018 with the role of the North Atlantic Ocean assuming importance in the recent
years. We next discuss the possible physical mechanism involved by which SST anomalies over key regions in
the North Atlantic influence the circulation over South Asia. Finally, CESM simulation results are used to
isolate the effect of North Atlantic SST variability on dust emission and transport over South Asia.
**3.1     Decadal change in correlation between dust and SST**
We have carried out correlation analysis of $DA_\%$ over the dust belt of South Asia with annual averaged SSTs
separately for the periods 2001-2010 and 2011-2018. These are the two periods when the signature of shift from
the Pacific to the Atlantic SST modulation of $DA_\%$ is the strongest. The maps showing spatial distribution of the
correlation coefficients for these two periods are shown, respectively, in Figures 2a and b. During 2001-2010,
the largest coherent region with which $DA_\%$ shows significant positive correlation encompasses central
equatorial Pacific (Fig. 2a; marked by continuous rectangle). During 2011-2018 this region has contracted and
shifted north-eastwards (Fig. 2b; continuous rectangle), while two new regions of significant correlations have
emerged: (1) over mid-latitude North Atlantic centered on 40$^o$N latitude (significant positive correlation) and (2)
over sub-tropical North Atlantic centered on 20$^o$N latitude off the western coast of North Africa (significant
negative correlation). These two regions are shown by dashed rectangles and are marked as "1" and "2"
respectively in Fig. 2b. Though a weak signature of this correlation pattern is present in 2001-2010, it has
emerged significantly strong during 2011-2018. Conducting month-by-month analysis of the impact of SST on
$DA_\%$ (not shown) it is seen that the positive correlation between $DA_\%$ and SST over central equatorial Pacific
during 2001-2010 is most prominent during September-October; while that over the North Atlantic during 2011-
2018 is most prominent during April-June, which are used here for subsequent analysis. We have constructed a
North Atlantic Difference Index (NADI) of SST by taking into account the regions where $DA_\%$ have significant
correlation with the North Atlantic SST as seen in Fig. 2b. NADI is the standardised difference in SST over
mid-latitude (Region 1, taken as 70$^o$W-25$^o$W longitude, 25$^o$N-40$^o$N latitude) and sub-tropical (Region 2, taken
as 70$^o$W-25$^o$W longitude, 10$^o$N-20$^o$N latitude) North Atlantic, averaged for April-June. Fig. 2c depicts the



variation of correlation coefficient between April-June NADI and monthly DA$_\%$ over South Asia separately for
2001-2010 and 2011-2018. Monthly DA$_\%$ is simply the percentage of days in a month when Ţ > 0.6 and α < 0.2.
Fig. 2c clearly shows that the correlation between NADI and DA$_\%$ is stronger and significant (at 95% confidence
level) for 2011-2018, during May-October, in comparison to 2001-2010. These months having significant
correlation largely coincide with the high dust months over South Asia, where dust loads peak during May-June.
During 2011-2018, conducting partial correlation analysis between April-June NADI and annual DA$_\%$ adjusted
for the central equatorial Pacific SST (taken as 178$^{o}$W-100$^{o}$W, 10$^{o}$S-10$^{o}$N) improves the correlation to 0.93,
which is significant at 99% confidence level. At the same time, partial correlation between the central equatorial
Pacific SST and DA$_\%$ adjusted for NADI gives a correlation coefficient of -0.36, which is not significant. For
2001-2010, a significant negative relation between NADI and DA$_\%$ is seen only for the month of February.

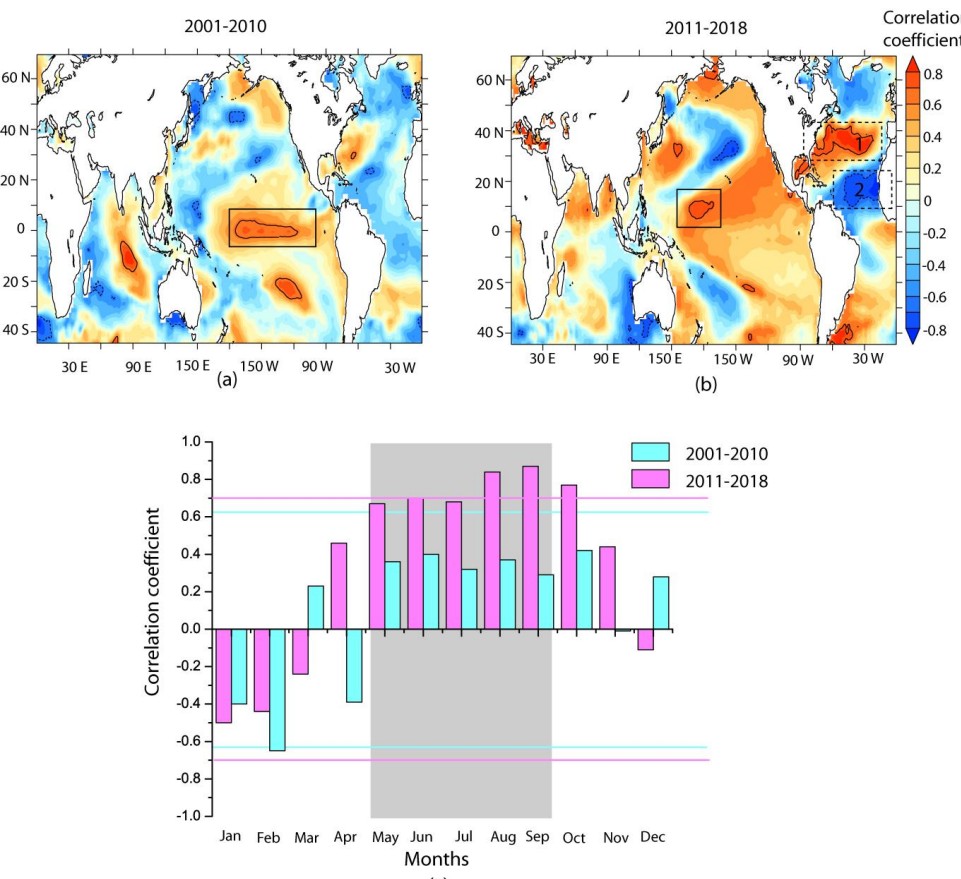

**Figure 2: Correlation between percentage frequency of annual dust activity (DA$_\%$) and annual average SST for (a)**
**2001-2010 and (b) 2011-2018. The black contours enclose the regions where correlation coefficient is significant at**
**95% confidence level. The continuous (dashed) boxes show the main regions with which DA$_\%$ over South Asia have**
**significant correlations over the Pacific (Atlantic) Ocean (see text for details). In (b) the regions used for constructing**
**the North Atlantic Difference Index (NADI) are marked as "1" and "2". (c) Correlation between April-June NADI**
**and monthly DA$_\%$ are plotted. The blue and pink horizontal lines indicate the 95% confidence levels for 2001-2010**





and 2011-2018 respectively. The grey shaded region highlights the months which have DA$_\%$ values greater than
annual average DA$_\%$.

The central equatorial Pacific, where the SST is significantly correlated with DA$_\%$ during 2001-2010, is
historically a prime region driving the variability of the SWM; and by some extension, dust emission and
transport. Several studies have shown that warming of the central equatorial Pacific SST leads to drought over
South Asia by inducing an anomalous descending motion (e.g., Kumar et al., 2006; Rajeevan and Pai, 2007).
Since the 1990s, stronger El Nino signals have been detected in the central Pacific SST compared to the eastern
Pacific (Yeh et al., 2009; Lee and McPhaden, 2010). Interestingly, there has been a cooling trend in the central
Pacific SST during 2001-2010 (of -0.8$^\mathrm{o}$C decade$^{-1}$) when this region was a major driver of DA$_\%$ over South Asia
(continuous box in Fig.3a). This formed a part of the hiatus within the ongoing global warming trend since the
beginning of the 21$^\mathrm{st}$ century, leading to a slowdown in global mean surface temperature warming rate to 0.02-
0.09$^\mathrm{o}$C (Xie and Kosaka, 2017). Several studies have shown that this has coincided with the negative phase of
the Pacific Decadal Oscillation and has been largely attributed to the internal variability over the Pacific Ocean
(Kosaka and Xie, 2013, 2016; Trenberth and Fasullo, 2013; England et al., 2014). The extreme El Nino of 2015
brought about the end of the global warming hiatus (Hu and Fedorov, 2017). This cooling trend is more
prominent during the boreal winter months (Trenberth et al., 2014).
With the end of the global warming hiatus, the North Atlantic Ocean emerged as an important driver of the
interannual variability of DA$_\%$ over South Asia during 2011-2018. A few recent studies have shown that since
late 1970s the Atlantic Ocean has assumed increasing influence over the climate of the Asian monsoon region as
the influence of the tropical Pacific has reduced (Kucharski et al., 2007; Sabeerali et al., 2019; Srivastava et al.,
2019). This in-turn impacts the circulation responsible for dust uplift and transport. The spatial pattern of
correlation between DA$_\%$ and SST for 2011-2018 in Fig. 2b shows resemblance to SST tripole pattern
associated with the positive phase of NAO (Bjerkness, 1964; Visbeck et al., 2001; Rodwell et al., 1999; Han et
al., 2016). In general, the positive phase of NAO projects to positive SST anomaly over the mid-latitude North
Atlantic and negative SST anomalies over the sub-tropical and the sub-polar North Atlantic (also see
Supplementary Fig. S1 a-c). DA$_\%$ is significantly correlated with the mid-latitude (Region 1 in Fig. 2b) and sub-
tropical (Region 2 in Fig. 2b) arms of the SST tripole. This tripole have recently changed sign from being
negative (warm phase) during 2001-2010 to positive (cold phase) during 2011-2018 (Supplementary Fig. S1 d-
e). That is, during 2011-2018, SST over North Atlantic shows a decreasing trend in the sub-tropics (centered on
20$^\mathrm{o}$N latitude), which is not significant, a significant (at 95% confidence level) increasing trend over the mid-
latitude (centered on 40$^\mathrm{o}$N latitude) and again a significant decreasing trend in the subpolar region (centered on
60$^\mathrm{o}$N latitude, dashed boxes in Fig. 3b). The SST trends over the North Atlantic during 2001-2010, on the other
hand, are not significant. In fact, December-February NAO index was neutral to negative during 2001-2010
(average NAO index -0.4) and changed to positive during 2011-2018 (average NAO index 2.4) (Delworth et al.,
2016; Iles and Hegerl, 2017) in tune with the switch in the sign of SST tripole during this period (Fig. 3c). Thus,
to sum up, with the resumption of global warming, the North Atlantic SST seems to assume importance in
controlling dust activity over South Asia, indicating a shift from the well-known importance of the Pacific SST.
The linkage is through a persistent positive phase of NAO during 2011-2018 and its imprint on the North





Atlantic SST tripole, the latter being in its positive (cold) phase during this period. In the next section we
discuss the physical mechanism responsible for North Atlantic SST leading to increased South Asian dust
activity.

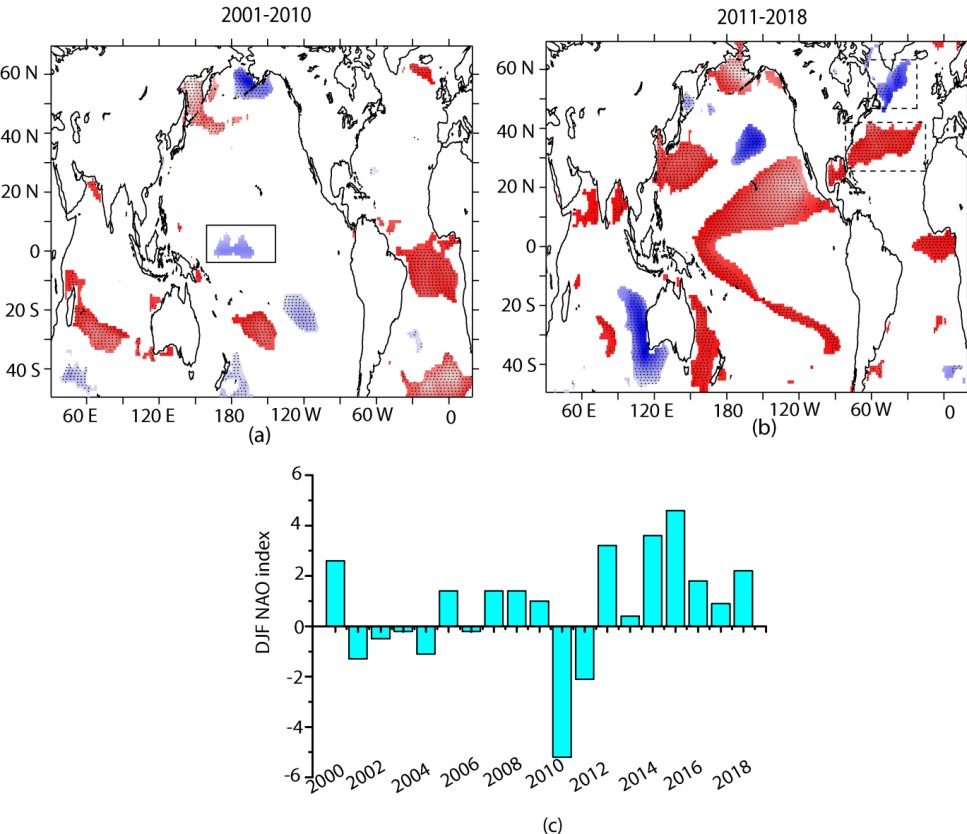


**Figure 3: Regions experiencing positive (red shades) and negative (blue shades) trends in sea surface temperature**
**during (a) September-October of 2001-2010 and (b) April-June of 2011-2018 significant at 90% confidence level. The**
**overlaid black stippling shows the regions where the trend is significant at 95% confidence level. (c) Time series of**
**December-February Hurrell's station-based NAO index for 2000-2018.**

**3.2    Physical Mechanism linking South Asian dust with Atlantic SST**
The above observations invoke the question: what could be the possible mechanism by which the changes in
North Atlantic SST impact South Asian dust activity during 2011-2018, when the Pacific Ocean influence has
reduced? The 'April to June North Atlantic Difference Index' (NADI, described in Section 3.1) is more strongly
and persistently correlated to winter and spring NAO index during 2001-2010 than during 2011-2018 (Fig. 4).
This indicates that the relation between winter and spring NAO and NADI (via SST tripole) has changed during
2011-2018, which has impacted circulation over South Asia and, thereby, dust load. To understand the
mechanism involved, we have estimated the correlation between April-June NADI and different meteorological





fields averaged for the months May-September when NADI is significantly correlated with DA$_{\%}$ (see Fig. 2c)
and also when high dust activity is widespread over South Asia. The results in Fig. 5 reveal that during 2001-
2010, NADI projects on to a cyclonic circulation anomaly at 850-700 hPa pressure level northwest off the
British Isles (red box in Fig. 5a) and a tripole-like SST anomaly with warming in the Norwegian Sea (Fig. 5b).
This resembles the Summertime East Atlantic (SEA) pattern, which is the second dominant mode of variability
after NAO over the North Atlantic Ocean during summer (Wulff et al., 2017; Osso et al., 2018; Osborne et al.,
2020), although, there are certain differences: (1) the cold sub-polar arm of the SST tripole has greater
southward extension (Fig.5b) and (2) an additional positive sea level pressure anomaly along western North
Atlantic between 10°N-50°N latitude is detected (Fig. 5c). Velocity potential at 850 hPa pressure level (green
contours in Fig. 5c) during May-September of 2001-2010 points to large-scale descending motion and
divergence over the North Atlantic. This is associated with negative precipitation anomalies over the cooler SST
regions of the North Atlantic, as well as, over Sahel (green contours in Fig.5b). The impact of NADI over South
Asia is mostly felt through the reduction in precipitation over west India and westerly anomalies in the south-
central Indo-Gangetic plain. Negative precipitation anomalies are also present over the dust source regions of
the Middle East and southern part of Central Asia.

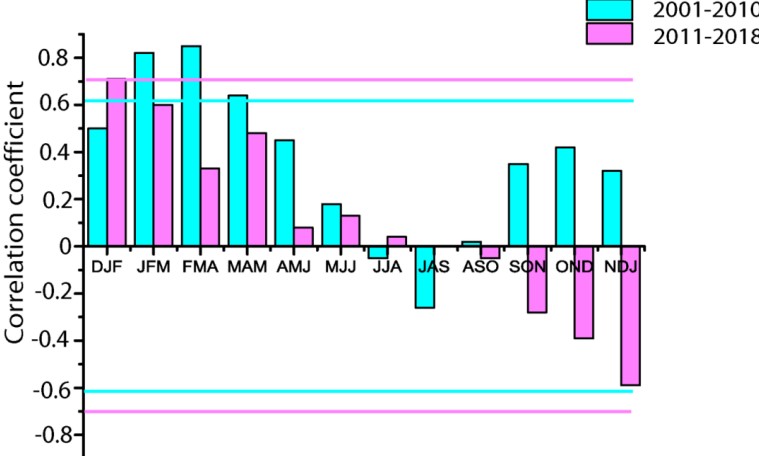


**Figure 4: Correlation between seasonal NAO index and April-June North Atlantic Difference Index (NADI)**
**separately for 2001-2010 and 2011-2018. The blue and pink horizontal lines indicate the 95% confidence levels for**
**2001-2010 and 2011-2018 respectively.**





During 2011-2018, the significant imprint of NADI on SEA wind pattern northwest off the British Isles is
absent (Fig. 5d), implying a shift in the relation between them. With the North Atlantic SST tripole changing
sign from warm during 2001-2010 to cold during 2011-2018 (Supplementary Fig. S1 d-e), NADI is significantly
correlated with the mid-latitude and sub-tropical arms of the SST tripole, but not with the sub-polar arm of the
SST tripole (shading in Fig. 5e). Additionally, there is an eastward shift in the region of positive correlation
between NADI and the mid-latitude arm of the tripole and a southward shift in the region of negative correlation



between NADI and the sub-tropical arm of the tripole. The region of low pressure off the British Isles, seen
during 2001-2010, is absent during 2011-2018 (Fig. 5f) due to the absence of the SEA pattern. Instead,
associated with the cooling of sub-tropical North Atlantic SST, a large region of positive correlation between
NADI and sea level pressure over the sub-tropical North Atlantic appears (Fig. 5f). These changes in relation
between NADI and the North Atlantic SST tripole have resulted in convergence, as indicated by 850 hPa
velocity potential (green contours in Fig. 5f), and positive precipitation anomaly over the Mediterranean region
including North Africa and northwestern part of the Arabian Peninsula (green contours in Fig. 5e). The
summertime wet anomaly over the Mediterranean region leads to anomalous descending motion over South
Asia, Middle East and East Africa, which is indicated by negative velocity potential at 850 hPa over this region
(Fig. 5f). The net effect is that the region of positive sea level pressure anomalies linked with the cooler sub-
tropical arm of the SST tripole now stretches to encompass the Sahel, Middle East, western India and the central
part of northern IO (orange shading in Fig. 5f). Over South Asia this development suppresses precipitation over
different regions of India and leads to general dryness. More importantly, as seen by the vectors in Fig. 5d, the
positive sea level pressure anomaly over the Middle East invigorates the westerlies carrying dust from
Southwest Asia to South Asia. The northerlies which are important for dusty weather over Pakistan-
Afghanistan-Iran are also strengthened.
In summary, although persistent positive phase of NAO prevailed during 2011-2018, a disassociation between
NAO and NADI influenced circulation over the Eurasian sector and over North Africa. Over South Asia and
surroundings, this projected to increased subsidence and positive anomalies of sea level pressure, which resulted
in general weakening of the monsoon and strengthening of the dust-transporting northerlies and westerlies.

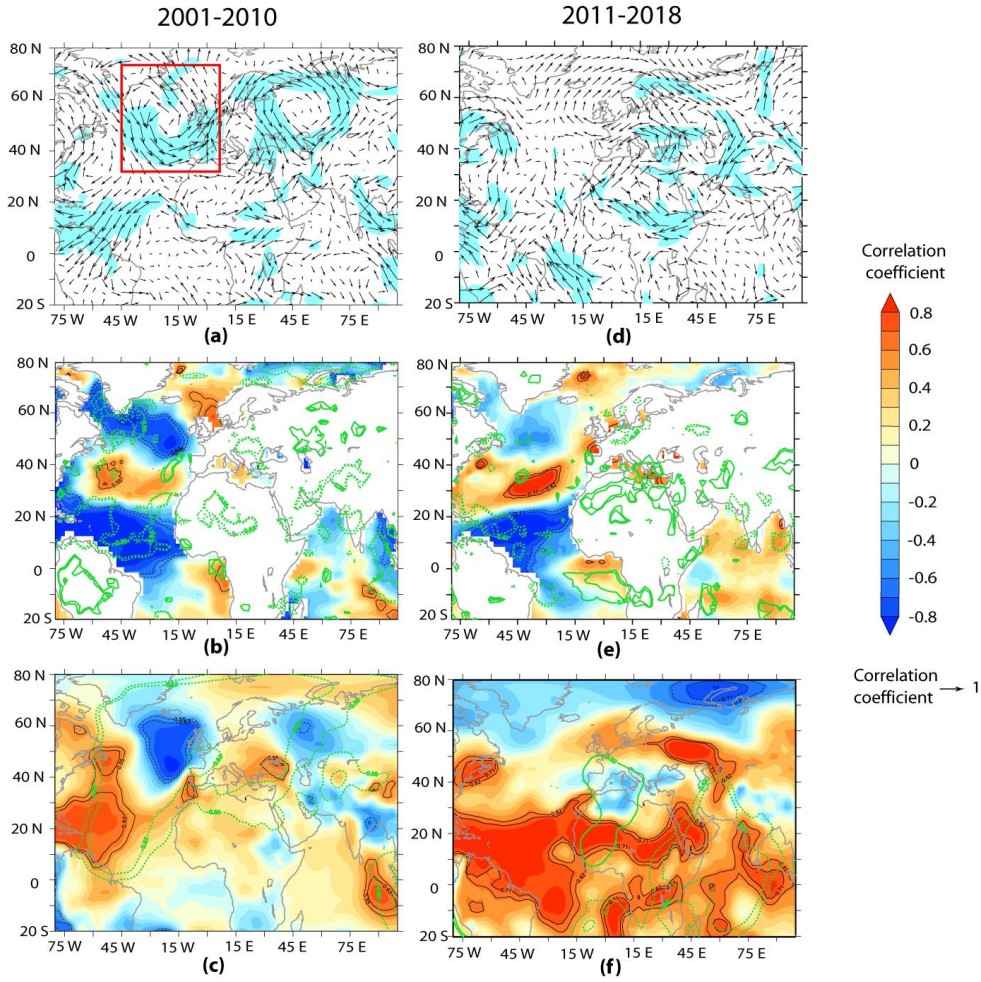


**Figure 5: Correlation between the April-June North Atlantic Difference Index (NADI) and different meteorological parameters from NCEP/NCAR Reanalysis averaged for May-September for (left panels) 2001-2010 and for (right panels) 2011-2018. (a) and (d) Arrows show correlation between NADI and wind vectors averaged between 850 and 700 hPa pressure levels. Light blue shade highlights the regions where one of the components of the wind vector is significantly (95% confidence level) correlated with NADI. (b) and (e) Shading shows correlation between NADI and SST and the green contours enclose the regions where significant correlation exists between NADI and precipitation. Black contours indicate the regions where correlation between NADI and SST are significant. (c) and (f) Shading shows correlation between NADI and sea level pressure and the green contours enclose the regions where significant correlation exists between NADI and velocity potential at 850 hPa pressure level. Black contours indicate the regions where correlation between NADI and sea level pressure are significant. For all the panels continuous and dashed contours are indicative of significant positive and negative correlations respectively; inner and outer contours of a particular colour indicate 95% and 90% confidence levels respectively.**

349

350

351



### 3.3     CESM simulation of Atlantic Ocean influence

The teleconnection between the North Atlantic SST and dust load over South Asia is explored further with the help of CESM simulations, with a view to isolate the contributions from North Atlantic SST anomalies. To achieve this, we have compared two sets of simulations, as explained in Section 2.2, for ten model years: one with climatological SST (Ctrl run) and the other with the SST trend for 2011-2018 superposed on the climatological SST over the North Atlantic (NAtl run). The difference (NAtl – Ctrl runs) yields the contribution solely from North Atlantic SST anomalies. It is important to note here that while NADI reflects the gradient between the mid-latitude and sub-tropical branches of North Atlantic SST, SST anomalies imposed for the NAtl run illustrate the response due to spatial pattern of SST anomalies over the entire North Atlantic due to persistent positive phase of NAO.

In general, CESM simulations can reproduce the main features of the North Atlantic summer climate and circulation, on which we are focussed here. Sea level pressure-based empirical orthogonal analysis carried out for CESM Large Ensemble simulations for 1920-2012 have revealed that NAO accounts for 40-member mean variance of 43% for winter months (Deser et al., 2017b). With our ten years CESM simulation we can still identify the dominant modes of variability. Empirical orthogonal function using June-September sea level pressure from CESM shows that NAO accounts for 63% and SEA pattern accounts for 14% of sea level pressure variances (Supplementary Fig. S2). To examine CESM performance over South Asia we have compared outputs from CESM Ctrl simulation with NCEP/NCAR wind at 850 hPa pressure level and PERSIANN precipitation separately for the spring inter-monsoon (April-May) and SWM (June-September) periods in Figs. 6 a-d. The comparisons reveal that the Ctrl run reproduces the main features of circulations and precipitation over South Asia fairly well, although with certain biases, which impact dust distribution and its temporal evolution. During April-May anomalous westerlies drive positive precipitation bias over peninsular India and southeast Bay of Bengal (Figs. 6 a and b). The anomalous southerlies over the southern part of the Indo-Gangetic plain lead to negative precipitation bias there, but a positive bias over the eastern Himalayas. During June-September, there are positive biases of precipitation along the west coast of India, southern India, the Himalayan foothills and most of the Middle East. Negative bias in precipitation prevails over eastern India and Southeast Asia bordering northeastern Bay of Bengal (Figs. 6 c and d). The positive bias along the west coast of India is associated with stronger westerlies in the Ctrl run. The anomalous anticyclone over the northern Bay of Bengal leads to a comparatively lower magnitude negative bias in precipitation of around 30%. This dipole in precipitation bias over the South Asian monsoon region has been recognized in Coupled Model Intercomparison Project Phase 5 (CMIP5) suite of models (Sperber et al., 2013) and has been attributed to several causes: SST bias over western equatorial IO (Annamalai et al., 2017); suppression of moist convection processes due to smoothening of topography (Boos and Hurley, 2013); weak advection of cold-dry air off Somali coast which reduces available moisture (Hanf and Annamalai, 2020). Comparing temporal evolution of CESM simulated precipitation with observations from PERSIANN (Figs. 6e and f) we see that generally wet bias prevails over both Indian domain (Fig. 6e) and the South Asian dust belt (Fig. 6f). CESM simulates one-month delay in the peak monsoon rainfall over these regions.



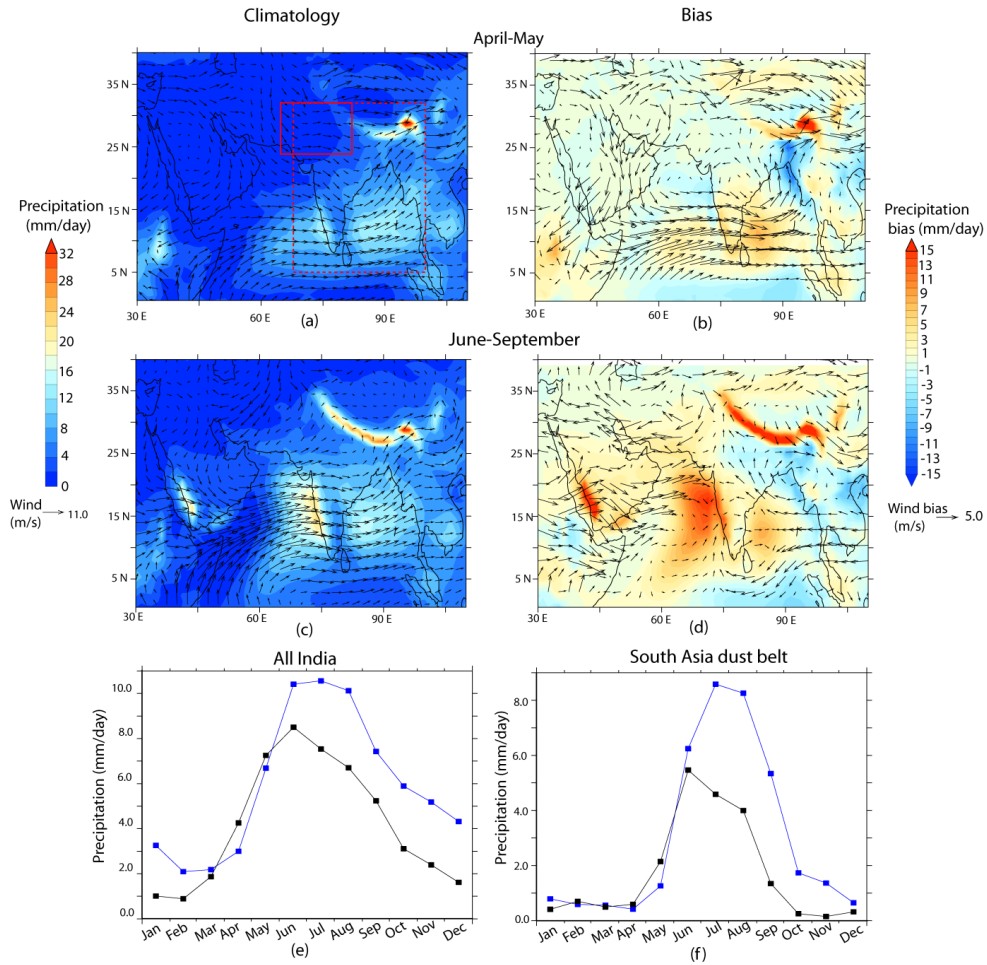

**Figure 6: Comparison of CESM-Ctrl simulation with observations/reanalysis data. CESM simulated climatology of precipitation and wind for (a) April-May and (c) June-September are shown. Differences between CESM simulated precipitation (shading) with that of PERSIANN and CESM simulated wind (arrows) with that of NCEP/NCAR reanalysis at 850 hPa pressure level are given for (b) April-May and (d) June-September. Time evolution of CESM (blue curve) and PERSIANN precipitation (black curve) over (e) All India (5°N-32°N latitude, 68°E-100°E longitude) and (f) the South Asian dust belt (24°N-32°N latitude, 65°E-82°E). These domains are, respectively, indicated in (a) by dashed and continuous red boxes.**

In general, CESM Ctrl reproduces the main dust emission regions over South and Southwest Asia (Fig.7a) along with temporal evolution of dust optical depth ($\tau_d$, Fig.7b). However, the positive bias in precipitation over dust source region, prevailing almost throughout the year, leads to underestimations of $\tau_d$ compared to observations. This discrepancy between CESM and observations is low during the winter months and increases during the monsoon months when CESM simulates about 3.5 mm day$^{-1}$ positive bias in precipitation over the South Asian dust belt and ~2 m s$^{-1}$ easterly wind bias. For example, during May when $\tau_d$ peaks, CESM simulates $\tau_d$ of ~0.2,



while AERONET coarse mode $\text{Ƭ}$ over Kanpur, Jaipur and Lahore are almost double. Negative bias in CESM $\text{Ƭ}_d$
is also apparent when compared to IASI-observed 10 μm $\text{Ƭ}_d$ over South Asia (Fig.7b). Although precipitation
bias during April-May is low (~0.1 mm day $^{-1}$, Fig. 6b), easterly wind bias of 0.7 m s$^{-1}$ leads to low transport
from the west. Similar negative bias in dust associated with weak northwesterlies over the Indo-Gangetic plain
has been noted for CESM-CAM5 simulation submitted to CMIP5 (Sanap et al., 2014). One important reason for
CESM underestimation of $\text{Ƭ}_d$ can be the exclusion of anthropogenic sources of dust, which contributes to nearly
half of the total annual dust emission (Ginoux et al., 2012). Several improvements in simulating dust with
CESM have been suggested by updating dust emission size distribution, optical properties, wet deposition
parameterizations and tuning soil erodibility (Albani et al., 2014). While further improvements in CESM for
better representation of dust cycle over South Asia is a topic for future, in case of this study, notwithstanding the
negative bias, CESM Ctrl simulation is able to simulate the pattern of spatial distribution and seasonal evolution
of South Asian dust. This is adequate for the present work as we are here interested in the direction of change in
simulated dust load due to the North Atlantic SST tripole rather than on the absolute magnitude of $\text{Ƭ}_d$. With this
understanding of the limitations of CESM simulation we proceed to examine the mechanism via which is SST
variability over the North Atlantic is responsible for perturbing dust load over South Asia.

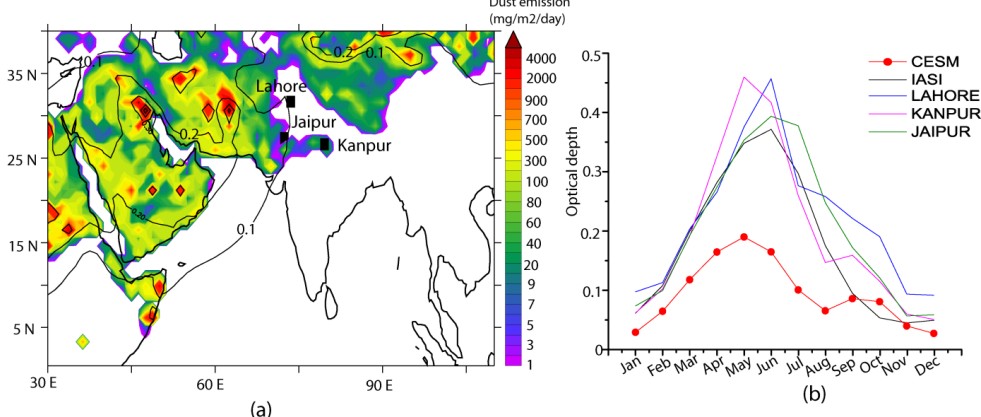

(a)              (b)


**Figure 7: (a) Shading shows the distribution of main dust emitting regions from CESM and the contours indicate dust optical depth. Both of these parameters have been averaged for ten model years. (b) Comparison of monthly climatology of dust optical depth from CESM-Ctrl simulation with IASI and AERONET (Lahore, Kanpur and Jaipur) observations.**


The differences between NAtl and Ctrl simulations for May-September are shown in Fig. 8, which highlights
that the North Atlantic SST anomaly, similar to during 2011-2018, can modulate South Asian dust activity via a
combination of reduced precipitation over the northern IO and strengthening of the dust-bearing northwesterlies
over the dust source regions. Cold SST tripole anomaly results in cooling in the upper troposphere and lowering
of the geopotential heights over South and Southwest Asia; both of which are important indicators of a weak
South Asian monsoon circulation (Fig. 8a). An east-west wave train over the mid-latitude and sub-polar region
of Eurasia sets-in with anticyclonic circulation over the sub-polar and cyclonic circulations over the mid-latitude





North Atlantic and also over the British Isles (Fig. 8b). Furthermore, a positive anomaly of sea level pressure
extends eastwards from the sub-tropical North Atlantic and is particularly strong over the northern IO. These
anomalies are similar to the response of the sea level pressure to NADI seen in the tropics; but are opposite to
that seen north of the mid-latitudes (Fig. 5f). Previously, model simulations have shown that the tropical North
Atlantic SST opposes the response of sea level pressure to the extra-tropical part of the cold SST tripole
(Osborne et al., 2020). A cyclonic circulation over the central equatorial IO and an anticyclonic circulation over
the northwestern IO inhibit the inflow of moisture into much of the Indian subcontinent leading to deficit
rainfall. It is the westerlies, which form the northern branch of the anticyclone, that transport dust from the
South Asian sources. For May-September, maximum increase in $\tau_d$ due to SST tripole is located over the South
Asian dust source region with dust emissions from the Thar being the main contributor (Fig. 8c). While over the
dust source regions the increase in $\tau_d$ is within 10%, dust transport by the strengthened westerlies can lead up to
20% increase in $\tau_d$ in the eastern Indo-Gangetic plain. Simultaneously, anomalous southerlies and
southeasterlies over the Arabian Peninsula suppress dust activity in the region (Fig 8b and c). The peak increase
in $\tau_d$ over South Asia due to North Atlantic SST is observed during June, when ~30% increase in $\tau_d$ compared
to CESM-simulated climatological values is achieved over the South Asian dust source regions (Fig. 8d). To test
the significance of the positive anomalies of $\tau_d$, we carried out Monte Carlo calculations by randomly selecting
6 years from NAtl and Ctrl simulations and differencing the $\tau_d$. By repeating this procedure 600 times, we find
that in 90% cases NAtl-Ctrl yields positive anomalies of $\tau_d$. It is important to note that although there is a
rainfall deficit over South Asia and the northern IO, only a small area within the main dust source regions are
impacted. This implies that a general increase in dryness and $\tau_d$ due to cold phase of North Atlantic SST tripole
is widespread over South Asia. However, the strengthened westerlies are responsible for enhanced dust flux
over the dust belt of South Asia. In this context, it is also worth mentioning that earlier works have reported that
cooling over the North Atlantic, either associated with the cold phase of Atlantic Multidecadal Oscillation or
due to the slowdown of the Atlantic Meridional Oscillation, is associated with weakened monsoon (e.g.,
Goswami et al., 2006; Zhang and Delworth, 2006; Feng and Hu, 2008; Liu et al., 2020). At decadal scale,
rainfall data for 1901-2004 showed that the positive (cold) phase of the SST tripole is associated with excess
monsoon over India due to strengthening of the westerlies over the northern IO (Krishnamurthy and
Krishnamurthy, 2015). However, the sign of correlation between the South Asian monsoon and the SST tripole
has undergone changes since 2000 with the negative (warm) phase of the SST tripole being associated with
strong monsoon over South Asia and vice versa (Gao et at., 2017), implying interdecadal shifts in the relation
between the two. These observations are supportive of our arguments above.

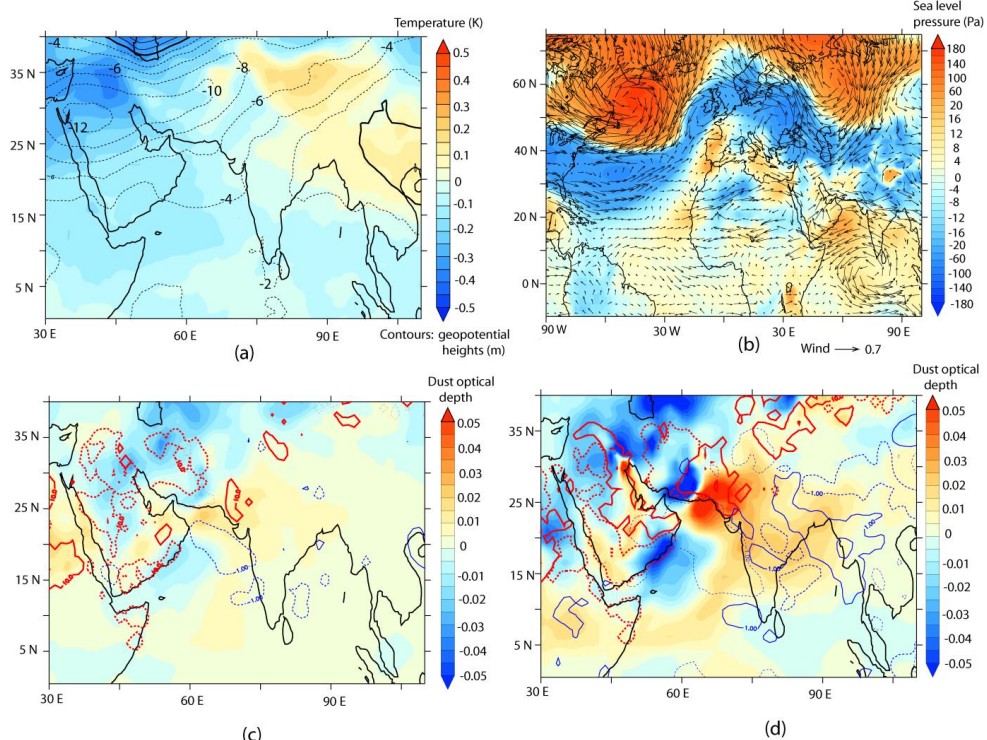

(c)                                    (d)

**Figure 8: Differences between CESM-NAtl and CESM-Ctrl simulations for (a-c) May-September. (a) Shading and contours indicate differences in temperature and geopotential height respectively at 200 hPa pressure level. (b) Shading indicates difference in sea level pressure and the arrows indicate difference in wind vectors at 850 hPa pressure level. (c) Difference in dust optical depth over the northern Indian Ocean and surrounding regions are shown by shading. The thick red contours enclose the regions where dust emission flux difference is greater than 10 mg m$^{-2}$ day$^{-1}$ and the thin blue contours enclose the regions where precipitation difference is greater than 1 mm day$^{-1}$. (d) Same as (c) but for the month of June. For all contours positive values are shown by continuous lines and negative values are shown by dashed lines.**

The increase in $\Upsilon_d$ discussed above is enabled by strengthening of dust-transporting westerlies at 800 hPa pressure level, which can, averaged for May to September, increase dust concentration by 20% at this altitude. This furthers when we analyse month-by-month changes in dust transport, as shown in Fig. 9, where a much stronger influence of North Atlantic SST tripole on dust concentrations is evident. The positive anomalies of dust concentration slowly start to build up during April to reach a peak during June and then subside by September. During May and June, the North Atlantic SST tripole can enhance dust concentration by 40-50% in the lower and mid-troposphere over the South Asian dust belt. These are also the months when maximum negative anomalies of precipitation are seen, following which positive anomalies of precipitation builds up. During May, maximum dust concentration anomaly centered on 800 hPa pressure level is associated with transport from the eastern Arabian Peninsula (due to anomalous southwesterly). During June, on the other hand, the strengthened northerlies transport dust all the way from eastern part of Central Asia into South Asia between 60°-75°E longitudinal belts. Additionally, descending motion above 500 hPa pressure level leads to trapping of





486    dust below this level. The overall weakening of the South Asian monsoon circulation is also demonstrated by

487    the anomalous upper level westerlies.

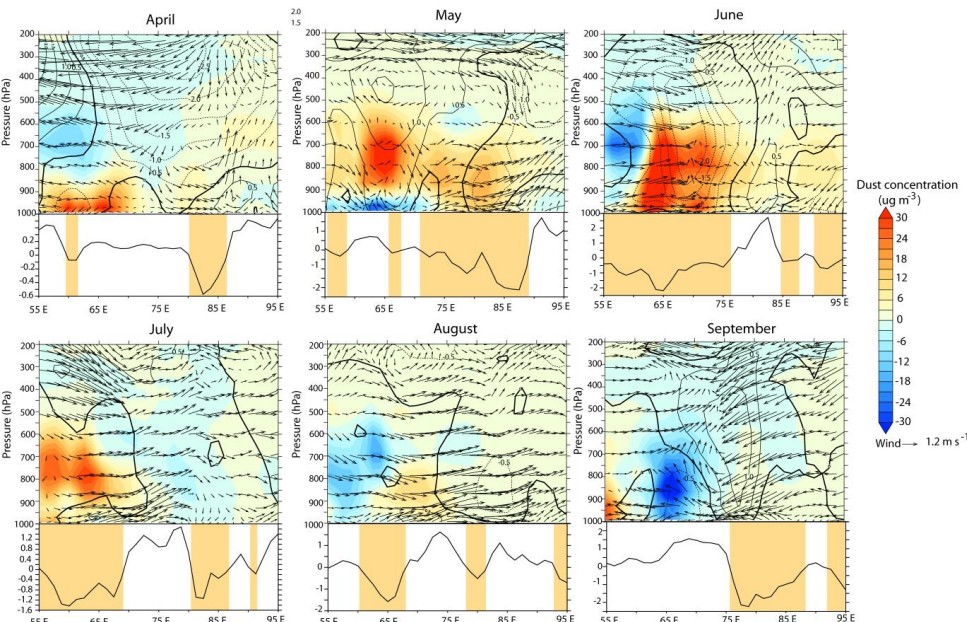

488

**Figure 9: Sections along 25°N latitude illustrating month-wise differences in dust transport between CESM-NAtl and CESM-Ctrl simulations. In upper part of each panel, shadings indicate difference in dust concentrations between the two simulations, the vectors are the differences in zonal and vertical components of wind and the contours are the differences in meridional component of wind. Continuous (dashed) contours indicate southerly (northerly) wind anomalies. The lower part of each panel plots precipitation differences, in mm day⁻¹, between CESM-NAtl and CESM-Ctrl simulations along 25°N latitude. The orange shades indicate the longitudinal belts which have negative anomalies of precipitation. Note that the vertical velocity is expressed as Pa s⁻¹ and has been multiplied by 40.**


**4        Conclusions**
Our study underlines the need to look at large-scale factors, which are global in nature, in significantly
modulating dust load over South Asia, in addition to changes in local factors. This is specifically relevant
considering the fact that about 50% of dust over this region is transported from remote (non-local) sources
(Banerjee et al., 2019). In this light, we have attempted to understand how changes in large-scale SST patterns
can impact dust emissions and transport pathways in this region. The "memory" of SST provides a bridge
between the circulation changes taking place across the globe. Our study relies on satellite data which are only
available since 2001. Even with this we see significant changes in terms of the relative importance of SST from
different regions driving interannual variability of dust over South Asia.
Our study shows that during the second decade of the 21ˢᵗ century the North Atlantic SST has emerged as a
dominant player in controlling dust activity over South Asia, in contrast to the hitherto important role played by
the Pacific SST. This is accompanied by the resumption of global warming following the early 21ˢᵗ century





warming hiatus and by persistent positive phases of NAO which has resulted in positive (cold) phase of the
North Atlantic SST tripole pattern. Specifically, high dust activity during 2011-2018 is associated with negative
SST anomaly over sub-tropical North Atlantic and positive SST anomaly over mid-latitude North Atlantic, the
two southern arms of the North Atlantic SST tripole. The difference in SST between these two arms of the
tripole, which we term as North Atlantic Difference Index or NADI, projects in to the SEA-like circulation
anomaly during May to September months of 2001-2010. Interestingly, during 2011-2018 a weakening of the
relation between NAO and NADI dilutes the impact of NADI on SEA. The result is a weakening of the South
Asian monsoon which leads to decreased precipitation and general increase in dryness with enhanced dust load.
Additionally, positive sea level pressure anomaly over South and Southwest Asia leads to anomalous northerlies
and westerlies which are responsible for transporting dust over South Asia. Sensitivity studies conducted with
CESM model shows that averaged for May-September the North Atlantic SST tripole anomaly can lead to
around 10% increase in dust optical depth, while it can contribute to 30% increase in dust optical depth during
the month of June. Most of the increase in dust load can be attributed to enhanced transport at 800 hPa pressure
level, which increases dust concentration by 20% for May-September and by as much as 40-50% during May-
June.
The present study demonstrates impact of the North Atlantic Ocean using 18 years of satellite data. However, in
the past, cold events in the North Atlantic have been associated with the slowdown of the South Asian monsoon
system and increase in dust fluxes over the northern Indian Ocean and Southwest Asia (e.g., Pourmand et al.,
2004; Mohtadi et al., 2014; Safaierad et al., 2020). Longer term data needs to be analysed from recent past to
better understand how this relation between dust and North Atlantic SST has fluctuated over time. This will
provide important clues as to how future relative changes in global SST in a warming world can control dust
fluxes over South Asia and the possible climate implications.

### Code availability

The code for CESM1.2 is available at https://www.cesm.ucar.edu/models/cesm1.2/

### Data availability

Level 3 MODIS Aqua+Terra version 6.1 daily aerosol data was downloaded from Level 1 and Atmosphere
Archive and Distribution System (LAADS) Distributed Active Archive Center (DAAC) website
(https://ladsweb.modaps.eosdis.nasa.gov/missions-and-measurements/science-domain/l3-atmosphere). IASI dust
optical depth was obtained from https://iasi.aeris-data.fr/dust-aod_iasi_a_data/. NCEP/NCAR meteorological
fields, NOAA ERSST version 5 data, OISST version 2, COBE SST version 2 data and GPCP version 2.3
precipitation data were obtained from National Oceanic and Atmospheric Administration (NOAA) Physical
Sciences Laboratory website (https://psl.noaa.gov/data/gridded/data.ncep.reanalysis.html). Monthly PERSIANN
precipitation data is maintained at University of California, Irvine (UCI), Center for Hydrometeorology and
Remote Sensing (CHRS) website (https://chrsdata.eng.uci.edu/). Hurrell's station-based NAO data is available
at        https://climatedataguide.ucar.edu/climate-data/hurrell-north-atlantic-oscillation-nao-index-station-based.
AERONET coarse mode aerosol data were obtained from https://aeronet.gsfc.nasa.gov/.



**Author contribution**

PB conceived the study, carried out model simulations, analyzed the data and wrote the manuscript. SKS and KKM contributed to scientific analysis and revision of the manuscript.

**Competing interests**

The authors declare that they have no conflict of interest.

**Special issue statement**

This article is part of the special issue "Interactions between aerosols and the South West Asian monsoon". It is not associated with a conference.

**Acknowledgements**

This research is supported by the Ministry of Earth Sciences (grant no. MM/NERC-MoES-1/2014/002). PB is also supported by Department of Science and Technology INSPIRE Faculty scheme. We acknowledge the computational facilities provided by Supercomputer Education and Research Centre (SERC) at the Indian Institute of Science for carrying out CESM simulations.

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
