# Peer review of "Is the Atlantic Ocean driving the recent variability in South"

_Atmospheric Chemistry and Physics, 2020_

## Author Comment (AC1)

**REPLY TO REVIEWER #1**

We thank the reviewer, Prof. Jérôme Brioude, for the constructive comments and suggestions, which have helped us to improve our manuscript. We have provided responses to the comments; the comments appear in *red italics*, our response in **bold face** below the respective comments and we have used *green italics* to quote the changes in the revised manuscript.

General comment: the paper presents new results on large scale processes that control the interannual variability of the dust concentration above the South Asia. The authors used satellite measurements from 2001 to 2018 to analyse the frequency of days (over a month, called DA%) when the dust optical depth above South Asia is high. Using NCEP/NCAR reanalysis, they found that an increase in DA% was associated to an increase in SST in the mid-latitude North Atlantic, and a cooling in the Subtropical North Atlantic between 2011 and 2018. The authors presented a detailed analysis, based on NCEP reanalysis and CEMS simulations, of anomalies in the wind fields and SST to explain the link between the SST variability in the Atlantic Ocean and the dust emission over South Asia. The correlation was linked to large scale transport pattern anomalies and a weakening of the South Asia monsoon.

The paper is well written and the results are of interest for the community. I will accept this paper for publication after addressing the following comments:

We thank the reviewer for the summary evaluation and positive recommendation. Our responses to the specific comments follow.

**Specific comment:**

 Section 3.3 is a bit long and probably needs some reorganisation. Figures 6 and 7 discuss the capabilities of CESM to simulate dust and precipitation in South Asia, and not so much the mechanisms that link dust activity to North Atlantic SST. I'm wondering if figure 6 and 7 should go in section 2.2 instead, and leave figures 8 and 9 in section 3.3. That way you will only discuss anomalies in section 3.3, which will help to follow the arguments of section 3.2.

This is indeed a helpful suggestion and we have complied with this. Figures 8 and 9, along with the relevant texts are now moved from Section 3.3 and placed under a separate Section 2.3 titled as "Model Validation". All the figure numbers in the manuscript are revised accordingly. The following sentences are added in the beginning of Section 3.3 to link it with the new Section 2.3:

As discussed in Section 2.3, although there are certain limitations, CESM can reproduce the main aspects of atmospheric circulation and the spatial and temporal characteristics of dust over South Asia quite well. This gives us confidence in using the model for our present study.

Please see L423-425 in the modified manuscript.

2) introduction: you don't mention the impact of the indian ocean dipole (IOD) and its impact on the monsoon and potentially on the dust emission. Please add references and comments.

Thanks for this suggestion. We have now added the following sentences in the introduction to elucidate the possible role of Indian Ocean Dipole on monsoon and dust:

The Indian Ocean Dipole (IOD) is the other teleconnection that influences atmospheric circulation over this region, with the positive phase of IODs counteracting the impact of El Nino on precipitation over South and Southwest Asia (Ashok et al., 2001; 2004). This can reduce the magnitude of anomalies of dust over Southwest Asia due to an El Nino event (Banerjee and Prasanna Kumar, 2016).

Please see L49-52 in the revised manuscript.

Technical comments:

figure 5 top: the coast lines need to be enhanced.

Complied with. Please note that the Figure number has been changed in the revised manuscript to Figure 7.

---

## Referee Report (RR1)

**Comments:**

The format and diagram of this article need to be modified. In addition, some problems and processes are unclear and need to be further explained. *Therefore, I recommend it be accepted for publication after major revisions.*

The serial numbers of all the following questions are dominated by the highlight version:

- Unify all longitude and latitude information formats in the text, such as line 192: "5°-80°N latitude and 5°W-85°W", line 366: "175-140°W".
- 2. Uniform row spacing, such as Figure 2,3,8.
- 3. Modify Figure 4, the icons are not aligned, and the figure needs to be redrawn.
- 4. Although the previous studies indicated that the NAO can impact North Atlantic tripole SST pattern, in your study, lines 395-398, the NAO has closely related to the NADI in the early period, when the Pacific SST is the dominant mode. However, in the latter period, the NAO is not related to NADI, when the North Atlantic SST is the dominant mode. Therefore, how does the NAO influence Indian dust through the North Atlantic tripole SST pattern? I am really confused this process.
- 5. Lines 412-413: We can't see how NADI affects South Asia in the distance. How does NADI affect precipitation in India? What is the process? Through remote teleconnection? Wave propagation or land surface process? It cannot be said that the Atlantic Ocean can affect India only through its local atmospheric circulation and precipitation.
- 6. Lines 633-634: How to explain Figure 7? There is no relationship between NAO and NADI?
- The format of references needs to be modified, such as line "Bjerknes, J:", line 794" 231-244", line 827" During El Ni no,", line 947" J. Geophys. Res.-Atmos.,",

8. "5. Line 197-198, why you divided two periods from 2010, what is the basis?

We have conducted several correlation analyses between DA% and SST by shifting the time window one year at a time. That is: 2001-2010, 2002-2011 and so on. This showed that the periods 2001-2010 and 2011-2018 have the most prominent signals of Pacific and Atlantic SST influence on dust, respectively. Hence, we followed this division of period."

It should be given that the sliding correlation between the DA% is and SST indices in the Pacific and Atlantic Ocean, and the transition point of time should be given by testing, rather than setting an artificial time window.

---

## Author Response (AR2)

**REPLY TO REVIEWER #2**

*Comments:*

*The format and diagram of this article need to be modified. In addition, some problems and processes are unclear and need to be further explained. Therefore, I recommend it be accepted for publication after major revisions. The serial numbers of all the following questions are dominated by the highlight version:*

We appreciate the valuable feedbacks from the reviewer, which have helped us to improve the manuscript. We have provided responses to the comments; the comments appear in *red italics*, our response in **bold face** below the respective comments and we have used *green italics* to quote the changes in the revised manuscript. Please note that, unless otherwise mentioned, all line numbers and figure numbers in our response are with respect to those of the revised manuscript.

*1. Unify all longitude and latitude information formats in the text, such as line 192: "5°-80°N latitude and 5°W-85°W" , line 366: "175-140°W".*

**Thanks for pointing this out. We have now expressed latitude and longitude information in uniform manner throughout the manuscript. Some examples are in L189, L229, L286 of the revised manuscript.**

*2. Uniform row spacing, such as Figure 2,3,8.*

**In this revised version of the manuscript all the figures have been thoroughly checked and modified to keep uniform spacing between each panel and uniform spacing of the texts in each of the figures.**

*3. Modify Figure 4, the icons are not aligned, and the figure needs to be redrawn.*

**Figure 4 has been modified to align all the icons. Also, based on the reviewer comment # 8 we have added an additional panel in this version of the manuscript (panel 4c) where we are showing 8-year running correlation between $DA_\%$ and April-June NADI and $DA_\%$ and the central equatorial Pacific sea surface temperature.**

*4. Although the previous studies indicated that the NAO can impact North Atlantic tripole SST pattern, in your study, lines 395-398, the NAO has closely related to the NADI in the early period, when the Pacific SST is the dominant mode. However, in the latter period, the NAO is not related to NADI, when the North Atlantic SST is the dominant mode. Therefore, how does the NAO influence Indian dust through the North Atlantic tripole SST pattern? I am really confused this process.*

**We want to bring to light in this study that 2 regions of the North Atlantic (that is, the 2 arms of NADI) are important in driving dust variability over South Asia. These two arms are located in the region influenced by the SST tripole pattern; the latter being associated with NAO. In that sense, there is an overlap between NADI and NAO. However, it is not the full picture. During different periods, there can be shifts in the relation between NAO and SST pattern and the magnitude of SST anomalies. This impacts the SSTs in the boxes considered for NADI and leads to divergences between NADI and NAO, as has been observed during 2011-2018. Consider for example, the following figure where we have compared NAO versus NADI impact**

on near-surface wind over South Asia. While there are some differences during 2001-2010, both NAO and NADI impacted wind over the central Indo-Gangetic plain and the Bay of Bengal. However, during 2011-2018, while NAO impacted the eastern end of South Asian dust source region, unlike NADI, NAO hardly had any impact on dust sources over southwest Asia. Thus, there are differences in how circulation over South Asia responds to NADI versus NAO between the 2 periods, which is why we focus in this study mainly on NADI impact on dust.

[Figure]

**Figure: Correlation between wind averaged over 700-850 hPa for May-September and (upper panels) December-February North Atlantic Oscillation index (NAO) and (lower panels) April-June North Atlantic Difference Index (NADI) for (left panels) 2001-2010 and for (right panels) 2011-2018. Light blue shade highlights the regions where one of the components of the wind vector is significantly (95% confidence level) correlated with NAO/NADI. Red stippling show dust emission regions obtained from CESM model simulations, and the red square encloses South Asian dust belt considered in this study.**

**To make this point clear, we have made edits throughout the manuscript as indicated below:**

**L14-18:** *Specifically, high DA$_{\%}$ is associated with warming in the mid-latitude and cooling in the sub-tropical North Atlantic SSTs: the location of the two southern arms of the North Atlantic SST tripole pattern. This shift towards a dominant role of the North Atlantic SST in controlling DA$_{\%}$ over South Asia coincides with a recent shift towards persistently positive phase of the North Atlantic Oscillation (NAO) and a resultant positive phase of the spring-time SST tripole pattern.*

**L331-332:** *DA$_{\%}$ is significantly correlated with the regions associated with these mid-latitude (Region 1 in Fig. 4b) and sub-tropical (Region 2 in Fig. 4b) arms of the SST tripole.*

**L341-343:** *However, as is shown in Section 3.3, during 2011-2018 there has also been a change in the relation between the North Atlantic SST anomalies, NADI and NAO, which remotely impacts circulation over South Asia.*

**L345-346: The sentence "***The linkage is through a persistent positive phase of NAO during 2011-2018 and its imprint on the North Atlantic SST tripole, the latter being in its positive (cold) phase during this period*" **has been changed to** "*This takes place during a persistent positive phase of NAO and positive (cold) phase of the North Atlantic SST tripole.* "

**L390-391:** *This indicates that, although persistent positive phases of NAO and SST tripole is observed, the relation between winter and spring NAO and NADI (via SST tripole) has changed during 2011-2018.*

**L448-450:** *In summary, although persistent positive phase of NAO prevailed during 2011-2018, a disassociation between NAO and NADI, via changes in the North Atlantic SST anomaly pattern, influenced circulation over the Eurasian sector and over North Africa.*

**L574-576:** *During this period, high dust activity is associated with negative SST anomaly over sub-tropical North Atlantic and positive SST anomaly over mid-latitude North Atlantic, the regions corresponding to the two southern arms of the North Atlantic SST tripole.*

*5. Lines 412-413: We can't see how NADI affects South Asia in the distance. How does NADI affect precipitation in India? What is the process? Through remote teleconnection? Wave propagation or land surface process? It cannot be said that the Atlantic Ocean can affect India only through its local atmospheric circulation and precipitation.*

**NADI impacts South Asia through: (1) modulating wavetrain propagation from the deep cyclonic circulation located to the northeast of the British Isles and (2) the extension of the anomalous surface pressure from the tropical North Atlantic into the northern Indian Ocean. While during 2001-2010, a prominent wavetrain emanating from the North Atlantic influenced Eurasia, the warming (cooling) of mid-latitude (sub-tropical) North Atlantic SST in 2011-2018 led to weakening of this wavetrain and positive sea level pressure anomalies over the tropical North Atlantic/Indian Ocean sector (see also Supplementary Fig. S3). The changes in wave amplitude and the location of the crests/troughs impact the northerlies and northwesterlies over the dust source regions, while the anomalous high pressure over the tropics leads to weakening of the South Asian monsoon. We have included the following lines in the revised manuscript to make the mechanism clear:**

**L401-405:** *The cyclonic circulation anomaly extends through the entire depth of the troposphere (not shown) and emanates wavetrain (Borah et al., 2020), as indicated by anomalies of the meridional wind in Fig. 8a. Anticyclonic circulation over Southwest Asia associated with the wavetrain translates to anomalous near-surface northerly over the central Indo-Gangetic Plain and the Bay of Bengal (Fig. 9a). This signals a weakening of SWM circulation.*

**L417-421:** *During 2011-2018, the significant imprint of NADI on SEA wind pattern northwest off the British Isles is absent, along with a weakening of the wavetrain (Fig. 8d), implying a shift in the relation between them. The anomalous southwesterlies over the eastern part of the Caspian Sea is replaced by anomalous northerlies. Near the surface level, this translates to stronger northerlies and northwesterlies over the dust source regions of southwest Asia and northwest India (Fig. 9b).*

**L444-447:** *Over South Asia this development suppresses the southwest monsoon circulation and precipitation over different regions of India and leads to general dryness. Together, a weakening of the southwest monsoon circulation and development of anomalous dust-bearing northerlies/northwesterlies over southwest Asia and northwest India drives an active dust season over South Asia.*

**L580-582:** *The result is a weakening of the South Asian monsoon circulation and development of anomalous dust-transporting northwesterlies and northerlies. This is facilitated by a weakening of the wavetrain from the North Atlantic and positive sea level pressure anomalies extending from the tropical North Atlantic into the northern IO.*

**We have furthermore modified Figures 8 a and d and have added Figure 9 to help explain wavetrain propagation and its impact on surface circulations over South Asia.**

[Figure]

**Figure 8: Correlation between the April-June North Atlantic Difference Index (NADI) and different meteorological parameters from NCEP/NCAR Reanalysis averaged for May-September for (left panels) 2001-2010 and (right panels) 2011-2018. (a) and (d) Arrows show correlation between NADI and wind vectors averaged between 500 and 200 hPa pressure levels significant at 95%. Red and blues shade highlight the regions where the meridional component of the wind has significant (95% confidence level) positive and negative correlations, respectively, with NADI.**

[Figure]

**Figure 9: Correlation between the April-June North Atlantic Difference Index (NADI) and wind averaged over 700-850 hPa for May-September for (left panels) 2001-2010 and for (right panels) 2011-2018. Light blue shade highlights the regions where one of the components of the wind vector is significantly (95% confidence level) correlated with NADI. Red stippling show dust emission regions obtained from CESM model simulations, and the red square encloses the South Asian dust belt considered in this study.**

*6. Lines 633-634: How to explain Figure 7? There is no relationship between NAO and NADI?*

**Please refer to our response to comment #4. We have additionally made few edits in the conclusion section to better convey the changing relation between NAO, NADI and the North Atlantic SST anomalies.**

**L572-581:** *From 2011 onwards, persistent positive phases of NAO resulted in positive (cold) phase of the North Atlantic SST tripole pattern. During this period, high dust activity is associated with negative SST anomaly over sub-tropical North Atlantic and positive SST anomaly over mid-latitude North Atlantic, the regions corresponding to the two southern arms of the North Atlantic SST tripole. The difference in SST between these two regions, which we term as North Atlantic Difference Index or NADI, projects into the SEA-like circulation anomaly and east-west wavetrain during May to September months of 2001-2010. Interestingly, changes in the pattern of the North Atlantic SST anomalies during 2011-2018 weakened the relation between NAO and NADI and also diluted the impact of NADI on SEA. The result is a weakening of the South Asian monsoon circulation and development of anomalous dust-transporting northwesterlies and northerlies.*

*7. The format of references needs to be modified, such as line "Bjerknes, J:", line 794" 231-244", line 827" During El Niño,", line 947" J. Geophys. Res.-Atmos.,",*

**We have modified the format of the references:**

*Bjerknes, J* **has been modified to** *Bjerknes, J.* **Please refer to L651 of the revised manuscript.**

**231-244: This reference has been corrected as:** *Hurrell, J. W., and Deser C.: North Atlantic climate variability: the role of the North Atlantic Oscillation, J. Marine Syst., 78, 28-41, https://doi.org/10.1016/j.jmarsys.2008.11.026, 2009.*

**Please see L730-731 of the revised manuscript.**

*El Niño* **has been corrected to** *El Niño* **in L763-764 of the revised manuscript.**

**Wherever applicable, we have corrected** *J. Geophys. Res.* **to** *J. Geophys. Res.-Atmos.* **In the revised manuscript. These are: L771, L775, L782, L793, L845, L861, L878 and L882.**

*8. "5. Line 197-198, why you divided two periods from 2010, what is the basis?*

*We have conducted several correlation analyses between DA% and SST by shifting the time window one year at a time. That is: 2001-2010, 2002-2011 and so on. This showed that the periods 2001-2010 and 2011-2018 have the most prominent signals of Pacific and Atlantic SST influence on dust, respectively. Hence, we followed this division of period."*

*It should be given that the sliding correlation between the DA% is and SST indices in the Pacific and Atlantic Ocean, and the transition point of time should be given by testing, rather than setting an artificial time window.*

**This has been complied with in the revised version of the manuscript. We have added a new panel to Figure 4 where we show 8-year running correlation between DA$_\%$ and NADI and DA$_\%$ and central equatorial Pacific sea surface temperature. This figure (shown below) shows the reduction in Pacific SST control on dust and increasing Atlantic control on dust, and thus justifies the splitting of the study period into 2001-2010 and 2011-2018.**

**We have added the following sentences (L285-289) in the revised manuscript to justify the chosen study periods:**

*Furthermore, Fig. 4c shows 8-year running correlations between DA$_\%$ over South Asia and September-October central equatorial Pacific SST (taken as 175$^o$W-140$^o$W longitude, 5$^o$N-5$^o$S latitude) and April-June NADI, which clearly demonstrates the transition from Pacific control of dust during 2001-2010 to North Atlantic control of dust during 2011-2018. This justifies our basis of separation of the two periods in Figs. 4 a-b.*

[Figure]

**Figure 4: (c) 8-year running correlation between DA$_\%$ and April-June NADI (black curve) and DA$_\%$ and the September-October central equatorial Pacific SST (blue curve). Horizontal axis shows the first year for each correlation window and the grey shaded regions mark the locations of 95% significant level.**